# The Administration of Resveratrol and Vitamin C Reduces Oxidative Stress in Postmenopausal Women—A Pilot Randomized Clinical Trial

**DOI:** 10.3390/nu16213775

**Published:** 2024-11-03

**Authors:** Araceli Montoya-Estrada, Aline Yunuen García-Cortés, José Romo-Yañez, Guillermo F. Ortiz-Luna, Arturo Arellano-Eguiluz, Aurora Belmont-Gómez, Vivian Lopéz-Ugalde, Guadalupe León-Reyes, Arturo Flores-Pliego, Aurora Espejel-Nuñez, Juan Mario Solis-Paredes, Enrique Reyes-Muñoz

**Affiliations:** 1Coordination of Gynecological and Perinatal Endocrinology, National Institute of Perinatology, Ministry of Health, Mexico City 11000, Mexico; araceli.montoya@inper.gob.mx (A.M.-E.); jryz@yahoo.com (J.R.-Y.); 2Human Reproductive Biology, National Institute of Perinatology, Ministry of Health, Mexico City 11000, Mexico; ialine.ygc@gmail.com; 3Peri and Postmenopause Clinic, National Institute of Perinatology, Ministry of Health, Mexico City 11000, Mexico; g_ortiz_luna@hotmail.com (G.F.O.-L.); artedro@hotmail.com (A.A.-E.); 4Coordination of Clinical Pharmacology, National Institute of Perinatology, Ministry of Health, Mexico City 11000, Mexico; aurobel89@yahoo.com.mx (A.B.-G.); lvivian_63@hotmail.com (V.L.-U.); 5Nutrigenetics and Nutrigenomics Laboratory, National Institute of Genomic Medicine, Ministry of Health, Mexico City 14610, Mexico; greyes@inmegen.gob.mx; 6Department of Immunobiochemistry, National Institute of Perinatology, Ministry of Health, Mexico City 11000, Mexico; arturofpliego@gmail.com (A.F.-P.); auro.espejel@gmail.com (A.E.-N.); 7Clinical Research Branch, National Institute of Perinatology, Ministry of Health, Mexico City 11000, Mexico; juan.mario.sp@gmail.com; 8Research Direction, National Institute of Perinatology, Ministry of Health, Mexico City 11000, Mexico

**Keywords:** oxidative stress, postmenopause, antioxidants, polyphenol, protein damage, total antioxidant capacity, malondialdehyde, insulin resistance

## Abstract

In postmenopausal women, due to endocrine changes, there is an increase in oxidative stress (OS) that predisposes them to cardiovascular and metabolic alterations. Sixty-one percent of women in this stage require a primary therapeutic strategy to decrease OS. This study aimed to evaluate the effect of resveratrol and vitamin C on OS in postmenopausal women. A randomized, double-blind clinical trial was carried out. Forty-six postmenopausal women with insulin resistance (HOMA-IR > 2.5) were included and divided into three treatment groups: group A: resveratrol, *n* = 13; group B: resveratrol + vitamin C, *n* = 15; and group C: vitamin C, *n* = 14. Between before and after the antioxidants, group B showed a decrease of 33% in lipohydroperoxides (*p* = 0.02), and malondialdehyde (MDA) decreased by 26% (*p* = 0.0007), 32% (*p* = 0.0001), and 38% (*p* = 0.0001) in groups A–C, respectively. For protein damage, group B is the most representative, with a decrease of 39% (*p* = 0.0001). For total antioxidant capacity (TAC), there were significant increases of 30% and 28% in groups B and C, respectively. For HOMA-IR, there were no significant differences among the study groups. Supplementation with this combination of antioxidants significantly decreases markers of OS in postmenopausal women. In addition, it increases TAC by up to 30%.

## 1. Introduction

Reproductive aging in women is characterized by a gradual decline in ovarian function that generally occurs after age 40 [1]. The decrease in the synthesis and secretion of estrogens is concomitant with anovulatory periods and alterations in the periodicity, duration, and intensity of menstrual bleeding. This initial stage is called perimenopause and is associated with a rapid increase in the amount and redistribution of abdominal fat [2]; it is characterized by the frequent presentation of neuroendocrine and vasomotor alterations that are expressed as muscle pain, sleep disturbances, hot flashes, depression, irritability, and urogenital atrophy, among others [3]. When menopause occurs and ovarian function decreases, a lower antioxidant capacity is evident. This adds to the concomitant progression of aging and increases the prevalence of components of the metabolic syndrome [4,5,6].

Fifteen percent of Mexican women are in menopause. In addition, approximately 6.4% of the population is currently over 60 years of age [7]. It is predicted that, with the increase in life expectancy, women will live for 80 to 85 years [7], in which case, they will spend almost a third of their lives in the postmenopausal state. For this reason, the prevalence of diabetes and cardiovascular diseases increases with postmenopause by up to 67% [8], which is correlated with oxidative stress (OS) and chronic inflammation. This imbalance between the antioxidant capacity and the generation of oxidant molecules gives rise to the process of OS [9,10], in which the predominance of oxidizing agents (reactive oxygen and nitrogen species: ERON ROS and RNS) can modify or induce biomolecule breakdown, such as lipids and proteins that make up cells and tissues, affecting homeostatic regulation and immunological efficiency [11]. Oxidative damage can increase comorbidities with different chronic degenerative diseases that worsen during menopause, postmenopause, and aging due to exacerbated OS [12].

Different therapeutic schemes with antioxidants have been used [13,14] to improve the antioxidant capacity in various processes. Antioxidants may have dichotomous roles in ROS production [15]. They are easily oxidized and can act as oxidants to induce damage when present in high concentrations. It is suggested that they be administered at least in pairs to avoid the generation of a new free radical that can behave as a pro-oxidant [15]. Our interest is in the use of resveratrol (3,5,4′-trihydroxystilbene), which is an antioxidant that belongs to the stilbene family; it is a natural phytoestrogen and a polyphenolic flavonoid. In its structure, it contains benzene groups, which gives it its antioxidant capacity [16]. There is no evidence of toxicity in humans with resveratrol doses of up to 2 g per day, so it can be consumed for its beneficial effects without apparent toxicity [17]. A wide range of therapeutic effects of resveratrol have been described, including cardioprotective, anti-sclerotic, and anti-inflammatory properties [18]; it has been reported to inhibit susceptibility to high-density lipoprotein cholesterol (HDL-C) oxidation, as well as platelet aggregation [19]. It reduces the concentrations of lipids and triacylglycerol and the synthesis of total cholesterol, apolipoprotein-B, and lipoprotein-A [20], while it increases the expression of glucose transporter type 4 (GLUT-4) and insulin sensitivity [21]. Resveratrol increases the activity of the antioxidant enzymes superoxide dismutase and glutathione peroxidase in the rat myocardium and aorta. It potentiates the expression of nuclear factor erythroid 2-related factor 2 (Nrf2) and reduces the mortality of mice exposed to catecholamine administration [22]. It reduces oxidative load through NADPH oxidase-mediated suppression of ROS and enhances the expression of several antioxidant enzymes [23,24].

Eicosanoids regulate pro-inflammatory processes that may be mediated by Sirtuin 1 (SIRT1), which, through deacetylation, suppresses NF-kB [25]. It increases the formation of nitric oxide with a vasculoprotective effect [26,27]. These effects of resveratrol on endothelial cells are promoted by the activation of the estrogen receptor and mitogen-activated protein kinase (MAPK), as well as the transcription factors FOXO1 and FOXO3A. It improves the synthesis of eNOS and the production of NO, as well as VEGF. It induced the expression of the vasculoprotective Kruppel-like factor 2 (KLF2) transcription factor by activating SIRT1 [27]. The effects of resveratrol include the regulation of insulin resistance [28,29].

There is a lack of studies with true statistical significance that show efficient antioxidant supplementation (e.g., resveratrol and vitamin C administered together, as proposed herein) that can promote antioxidant synergism, considering an in vitro system. In this context, to avoid the depletion and oxidation of circulating reduced glutathione (GSH) by oxidized resveratrol, the participation of an electron donor molecule is necessary, such as vitamin C, which acts as a powerful antioxidant [30]. This study aimed to evaluate the effect of the administration of resveratrol and vitamin C on OS in postmenopausal women.

## 2. Materials and Methods

### 2.1. Study Design and Participants

This study was a randomized, double-blind clinical trial conducted in the Peri- and Postmenopause Clinic of the Instituto Nacional de Perinatología (INPer) in Mexico City from January 2019 to March 2020. It was carried out following the principles of the Declaration of Helsinki and was approved by the Internal Review Board of INPer, registered under the number 3210-10209-01-574-17. This clinical trial was registered in the International Registry of Clinical Trials, “clinicaltrials.gov”, with the number NCT03090997. Each participant signed an informed consent form.

The inclusion criteria were women between 50 and 60 years who were diagnosed with early postmenopause according to the Stage of Reproductive Aging Workshop (STRAW) classification (+1a, +1b, and +1c) and presented insulin resistance, as defined by the homeostatic model for assessing insulin resistance (HOMA-IR) ≥ 2.5, which was obtained from the following arithmetic operation: [fasting glucose mg/dL] * [fasting insulin mcUI/L]/405 [31,32]. Women with some alteration in their lipid profile were only included if they did not request pharmacologic treatment (total cholesterol < 280 mg/dL, triglycerides < 300 mg/dL, and fasting glycemic level <110 mg/dL).

The exclusion criteria were women with indicated hormone replacement therapy; women who were taking anticoagulants, metformin, bezafibrate, statins, or any antioxidant in the three months before study entry; women who had a diagnosis of diabetes mellitus, rheumatoid arthritis, lupus, neoplasms of any type, HIV, or kidney disease; active smokers; and women who did not agree to participate in the study.

Women who did not comply with 80% adherence to treatment were considered lost to follow-up and excluded from the study.

### 2.2. Randomization and Allocation Concealment

Participants were randomly assigned to one of three antioxidant treatments: resveratrol and a vitamin C placebo (group A), resveratrol and vitamin C (group B), and vitamin C and a resveratrol placebo (group C). The food supplements used were resveratrol capsules 500 mg (Resvitále; GNC, Brookhaven, PA, USA) and vitamin C/ascorbic acid tablets (C-500; GNC, Brookhaven, PA, USA). The placebos used were fabricated according to good practices for manufacturing by Carnot laboratories^®^, Mexico City, Mexico; for group A, the placebo was a tablet with the same presentation, size, and color as vitamin C, and for group C, it was a capsule with the same presentation, size, and color as resveratrol. Pharmacologists from the institute carried out the packaging of the antioxidants and placebos in opaque bottles with the same presentation, labeled with letters according to groups A, B, and C.

The treatment for each participant was assigned randomly using envelopes containing the letter of the treatment group. Subsequently, each participant received two bottles labeled with the letter of the assigned group. Neither the researcher nor the participant knew the contents of each bottle; only the pharmacologist, who did not participate in the assignment or delivery of the treatment, knew the contents of each bottle.

The research team and participants were blinded to the group allocation until the final analysis. The treatment interventions in both groups were similar in terms of packaging presentation and capsule shape and color. Patients were instructed to take the assigned antioxidants once a day before breakfast and to return after 30 days to receive a new bottle. Treatments were provided for three months. Participants were instructed not to consume any antioxidant supplements during the study and to continue their habitual diet.

### 2.3. Primary Endpoint

The primary endpoint was the effect of the co-administration of resveratrol and vitamin C in reducing markers of OS, which were evaluated in three groups. Blood samples were obtained by venipuncture before and after three months of supplementation with resveratrol, vitamin C, or an antioxidant placebo. The clinical status was monitored, and variables such as waist measurement, blood pressure recording, and laboratory studies that included glucose, triglycerides, total cholesterol, HDL-C, and low-density lipoprotein cholesterol (LDL-C) were recorded.

### 2.4. Secondary Outcomes

The secondary outcomes were the effect of the antioxidants in each group intervention in reducing insulin resistance in postmenopausal women. The insulin resistance was defined by the homeostatic model assessment of insulin resistance (HOMA-IR) >2.5.

### 2.5. Blood Sample

Whole-blood (6 mL) samples were obtained from each participant via an antecubital venipuncture into anticoagulant tubes with 2 mM ethylenediaminetetraacetic acid (BD Vacutainer, Franklin Lakes, NJ, USA). The samples were centrifuged at 3000 rpm for 15 min to obtain the plasma and were stored in aliquots at −70 °C until their analysis.

Two blood samples were taken: the first, called basal, was taken before any treatment, and the second at the end of the third month to obtain the values of glucose, insulin, lipid profile, markers of OS, and antioxidant capacity. At the end of the study, the patients continued with their usual climacteric follow-up.

### 2.6. Markers of OS: Biochemical Analysis

To detect oxidative damage, the following methodologies were quantified in plasma from postmenopausal women.

Determination of lipohydroperoxides. The propagation stage was determined based on the oxidative activity of lipohydroperoxides, which convert iodide to iodine, according to the method described by El-Saadani. The absorbance was determined at 360 nm [33].

Determination of malondialdehyde (MDA). MDA is one of the end products of lipoperoxidation. The absorbance was determined in a spectrophotometer at 584 nm, and tetraethoxypropane (TEP) was used as a standard solution [34].

Determination of protein carbonylation. Protein carbonylation was analyzed spectrophotometrically at a wavelength of 370 nm [35].

Determination of the total antioxidant capacity. The total antioxidant capacity in plasma was determined according to the CUPRAC method. The reduction of the cupric ion was estimated spectrophotometrically at 450 nm [36]. All experiments were performed in duplicate. Unless otherwise specified, all reagents used in this study were purchased from Sigma Chemical Co. (St. Louis, MO, USA). The food supplements used were resveratrol 500 mg (Resvitále; GNC, USA) and vitamin C/ascorbic acid tablets (C-500; GNC, USA).

### 2.7. Sample Size

Because this was a pilot study, a convenience sample of 15 participants per group was selected.

### 2.8. Statistical Analysis

Statistical analysis was performed using the Prism 8.0 program (GraphPad, San Diego, CA, USA). Continuous variables were expressed as mean ± standard deviation, and one-way analysis of variance (ANOVA) with Bonferroni correction or the Kruskal–Wallis test was used to compare inter-group continuous variables. Additionally, we performed an intra-group analysis using the paired Mann–Whitney U test. Statistical significance was set to *p* ≤ 0.05. The analysis of normality was performed using the Kolmogorov–Smirnoff test.

## 3. Results

Of the 80 women who attended the invitation to participate, 29 declined for personal reasons. Fifty-one women met the inclusion criteria and were allocated to an intervention. Nine women were lost to follow-up. Initially, 15 women were allocated to the intervention; however, in group B, 6 women withdrew their participation in the first two weeks of the intervention and were replaced, so in group B, a minimum of 15 women were allocated to the intervention. A total of 42 finished the follow-up (group A: resveratrol, *n* = 13; group B: resveratrol and vitamin C, *n* = 15; and group C: vitamin C, *n* = 14) (Figure 1).

Table 1 shows the comparison of basal clinical and biochemical characteristics among the three groups. Three months after the intervention, there were no significant differences in the basal clinical and biochemical characteristics among groups. After three months of intervention, there was a significantly lower concentration of total cholesterol but a significantly higher concentration of triglycerides in group C versus group A. Likewise, there was a significantly lower concentration of triglycerides in group B versus group C.

Table 2 shows the intra-group comparison of basal clinical and biochemical characteristics versus three months after the intervention. No significant differences were observed between the basal values and the values three months after the intervention for weight, BMI, glucose, insulin, lipid profile, and uric acid in each group of the study.

Figure 2 shows the comparison among groups; there were no significant differences in values at baseline and after three months of treatment. However, in the intra-group comparison between baseline and after three months of treatment, in the resveratrol + vitamin C group, there was a significant decrease of 33% in lipohydroperoxides (LPHs) (0.039 ± 0.015 vs. 0.026 ± 0.014 nmol LPH/mg dry weight, *p* = 0.03). There were no significant differences in groups A and C (*p* = 0.10 and *p* = 0.18, respectively).

Figure 3 shows the group comparison of MDA concentrations. The three groups presented statistically significant differences between baseline and after three months of treatment. The resveratrol group presented a significant decrease of 26% (*p* = 0.0007), the resveratrol + vitamin C group showed a decrease of 32% (*p* = 0.0001), and the vitamin C group had a decrease of 38% (*p* ≤ 0.0001) in MDA. The concentrations are expressed as nmol MDA/mg dry weight.

Regarding oxidative damage to proteins, Figure 4 shows the comparison between baseline and three months after the intervention; the three groups show statistically significant differences, the most representative being the resveratrol + vitamin C group with a decrease of 39% (17.41 ± 1.30 vs. 10.52 ± 1.20 nmol PC/mg of protein, *p* ≤ 0.0001), followed by groups A and C with a decrease of 29%.

On the other hand, when evaluating the total antioxidant capacity, expressed as nmol of Trolox equivalent/mg of protein (Figure 5), a statistically significant increase is observed when comparing the pre- and post-treatment values in the groups. The combination of antioxidants presented an increase of 30% (*p* = 0.001), and the vitamin C group had an increase of 28% (*p* = 0.03).

Finally, for insulin resistance (Figure 6), the HOMA-IR index did not show statistically significant differences in any of the three groups.

## 4. Discussion

In the present pilot study, supplementation with resveratrol and/or vitamin C alone or in combination for three months in postmenopausal women did not show significant differences in insulin resistance among the three groups in the intra- and inter-group comparisons. Even though there were significant decreases in LHP, MDA, and carbonylation after supplementation in the three groups compared to before the treatment in the intra-group analysis, none of the three interventions were superior to the others.

In 2020, our work group reported that postmenopausal women present higher concentrations of OS markers than women of reproductive age [37]. Therefore, using antioxidants can be a tool to reduce markers of oxidative damage to lipids and proteins. Recent studies have reported that resveratrol and vitamin C have beneficial effects, as cardioprotective, anti-sclerotic, anti-inflammatory, and antioxidant properties stand out [16].

In our study, in the intra-group comparison, we identified a statistically significant decrease in the lipoperoxidation process (propagation stage) with the use of antioxidants, and in the inter-group comparison, we observed that the decrease with the combination of resveratrol and vitamin C was greater (33%) than with resveratrol (25%) or vitamin C (15%); however, it did not reach statistical significance, probably due to the limited sample size.

Recently, Zhang et al. [38] demonstrated, in an experimental cell culture model using the MIN6 β-cell line, that resveratrol in pancreatic β cells decreases LPHs and one of the final products (acrolein). Likewise, Toaldo et al. [39] demonstrated that consuming grape juices rich in polyphenols promotes decreased lipoperoxidation in healthy Brazilian people. However, no studies have shown beneficial results of the combined effect of antioxidants in the human population.

In the last stage of lipoperoxidation, the final product, MDA, decreased in the three treatment groups (A, B, and C), with a statistically significant difference in the intra-group comparison. This decrease was similar to the results of studies carried out in 2019 and 2021 by Lin et al. [40] and Kong et al. [41] in an experimental animal model (Sprague-Dawley male rats), to which between 20 and 40 mg/kg/day of resveratrol was administered for a period of 15 days, showing a 33% decrease in MDA quantification. Regarding the combination of antioxidants, our study presented a reduction of 26% (*p* = 0.0001). Concerning the group with vitamin C, a reduction of 38% was also observed (*p* < 0.0001). This is the one with the highest percentage. Similar data in a population of 60 postmenopausal women with breast cancer demonstrated that the co-administration of vitamin C is more beneficial by reducing the levels of thiobarbituric acid in breast cancer patients treated with tamoxifen [42]. A longitudinal study carried out in a mixed population of subjects, men and women under 18 years of age, showed that 200 mg/day of vitamin C for 15 days reduces MDA concentrations by up to 52% [43].

Continuing with the chain of events of oxidative damage, we quantify the damage to proteins through the exposure of carbonyl groups. Our study showed a significant decrease in all three groups. However, the combined treatment of resveratrol and vitamin C showed a more significant effect, with a decrease of 39% (*p* < 0.0001). There is little evidence to support our findings; however, from what was obtained, we can propose that the combination of antioxidants has a beneficial effect in reducing oxidative damage [43].

Regarding antioxidant defense, we evaluated the total antioxidant capacity in plasma, demonstrating an increase of 11% with the use of resveratrol. Similar data were reported by Movahed et al. [44], showing an increase in TAC of approximately 35%. Hosseini et al. [45] showed that the intake of resveratrol at 500 mg per day for a period of 4 weeks in subjects with type 2 diabetes mellitus (DM2) increases TAC by 58% because its structure has benzene groups, which gives it an antioxidant capacity [16]. On the other hand, we can observe a greater increase when resveratrol is combined with vitamin C (30%). However, the use of vitamin C supplementation had a greater effect on TAC at 28%, a result contrary to ours, when Amini et al. [46] administered 1000 mg/day for 8 weeks in 60 women with endometriosis. According to our results, we suggest that vitamin C has a favorable impact on the increase in TAC due to its ability to donate electrons in its conversion to the ascorbic radical or dehydroascorbic acid, with its final reduction to ascorbic acid at the mitochondrial level [30].

Several studies support the concept that the oxidative state could play an essential role in insulin resistance since, at the muscle level, it contributes to the development of diabetes mellitus, which is associated with the overload of nutrients such as free fatty acids, glucose, and amino acids. Significant changes were not shown in any of the three groups regarding insulin resistance measured by HOMA-IR. Data similar to our findings were reported in 2020 by Thaung et al. [47], who administered 150 mg/day of resveratrol for 12 months in postmenopausal women without finding a decrease in HOMA-IR. Movahed et al. [44], in an experimental clinical trial with an Iranian population of 13 patients with type 1 diabetes, demonstrated that the administration of resveratrol 500 mg two times a day for two months did not present positive effects on HOMA-IR. Asghari et al. [48] showed no significant difference in using 600 mg of resveratrol daily for 12 weeks in subjects with fatty liver. Likewise, in 2014, Bashmakov et al. [49] reported that 100 mg per day for two months has no significant effect on insulin resistance. Similar to the results of the present study, we did not find a statistically significant difference in HOMA-IR, contrary to Abdollahi et al. [50], who reported that in a population of subjects with type 2 diabetes mellitus, with a mean age of 50 years, a dose of 1 g per day for eight weeks of resveratrol improved insulin resistance (*p* = 0.01).

This article has some great strengths. It is the first pilot study that evaluates the combination of two antioxidants and its effect on OS and insulin resistance. Another strength of the study is that it only included women in postmenopause STRAW +1a, +1b, and +1c, which is the period in which the most important hormonal, metabolic, and OS changes occur.

This pilot study has limitations, including its use of only three groups without a control group (without antioxidant treatment). However, comparing one, the other, or the combination of antioxidant supplements before and after treatment allowed us to observe the differences between the treatments applied in each group. Women were not left without any intervention, with the following justification: the effect of the combination of antioxidants (new treatment) was compared with the group that only received a single antioxidant to see whether the new treatment works to a greater extent due to a possible synergism or potentiation of effects by the sum of two antioxidants. In addition, the study lacked a control such as estrogen, estradiol, or follicle-stimulating hormone concentration. Another limitation was the inclusion of only women from the metropolitan area of the Valley of Mexico, which limits the generalization of the results. Therefore, our findings should be interpreted with caution.

## 5. Conclusions

Supplementation with resveratrol and vitamin C significantly decreases lipohydroperoxides and carbonylation. Vitamin C increased the total antioxidant capacity of postmenopausal women with insulin resistance by up to 33%. However, using resveratrol and/or vitamin C alone or in combination did not present significant differences in insulin resistance. Therefore, it is suggested that a study be carried out with an appropriate sample size.

## Figures and Tables

**Figure 1 nutrients-16-03775-f001:**
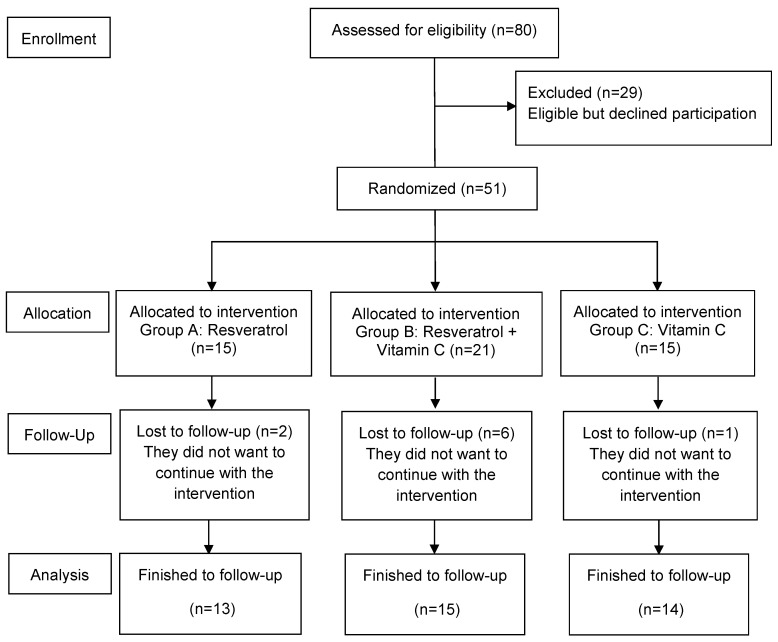
Flowchart of the participants in the study.

**Figure 2 nutrients-16-03775-f002:**
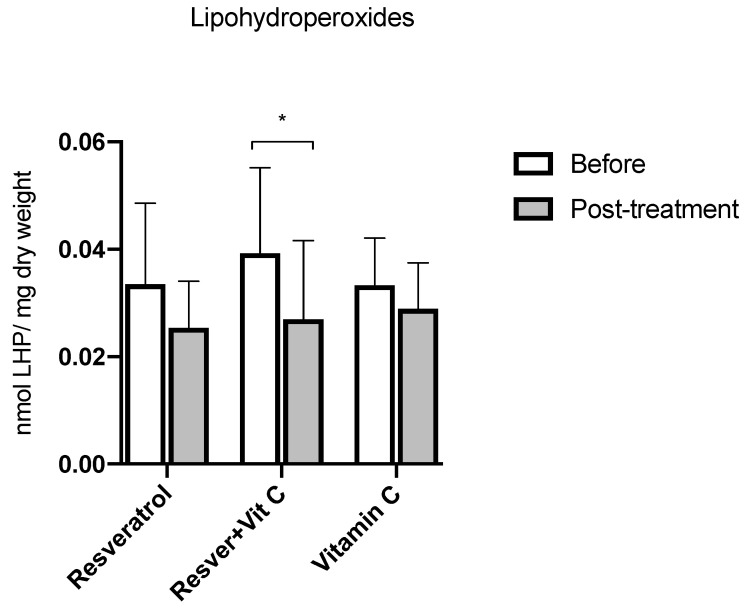
LHP concentration before and after treatment in postmenopausal women with insulin resistance. Resveratrol group, *n* = 13; resveratrol and vitamin C group, *n* = 15; and vitamin C group, *n* = 14. The results obtained were analyzed using Student’s “t”. Data are presented as mean ± standard deviation. * *p* < 0.05.

**Figure 3 nutrients-16-03775-f003:**
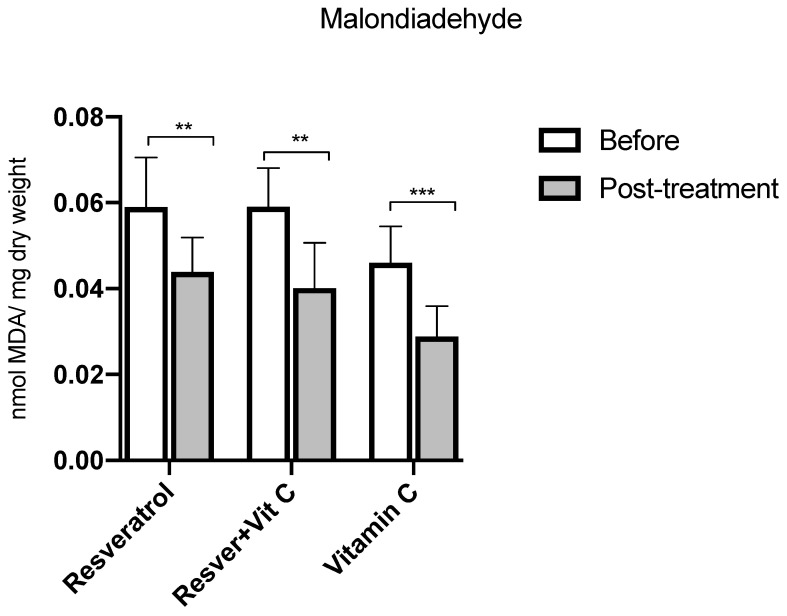
MDA concentrations before and after treatment in postmenopausal women with insulin resistance. Resveratrol group, *n* = 13; resveratrol and vitamin C group, *n* = 15; and vitamin C group, *n* = 14. The results obtained were analyzed using Student’s “t”. Data are presented as mean ± standard deviation. ** *p* < 0.01, *** *p* < 0.001.

**Figure 4 nutrients-16-03775-f004:**
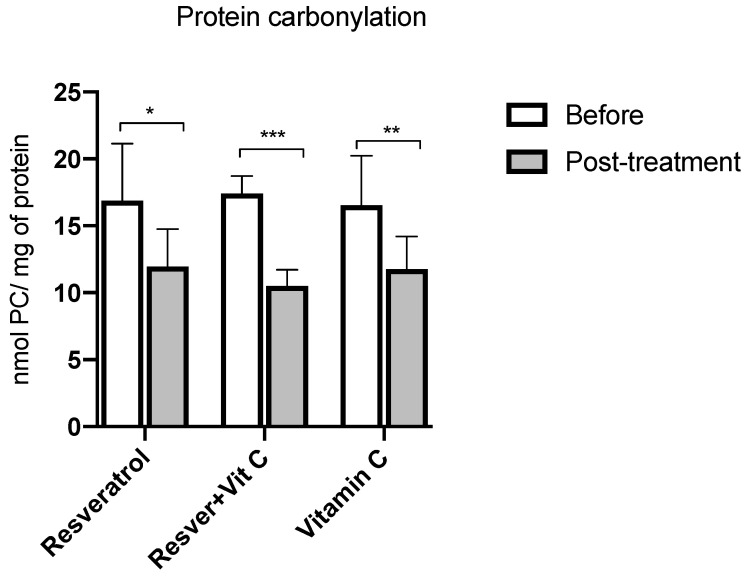
Protein carbonylation concentration before and after treatment in postmenopausal women with insulin resistance. Resveratrol group, *n* = 13; resveratrol and vitamin C group, *n* = 15; and vitamin C group, *n* = 14. The results obtained were analyzed using Student’s “t”. Data are presented as mean ± standard deviation. * *p* < 0.05, ** *p* < 0.01, *** *p* < 0.001.

**Figure 5 nutrients-16-03775-f005:**
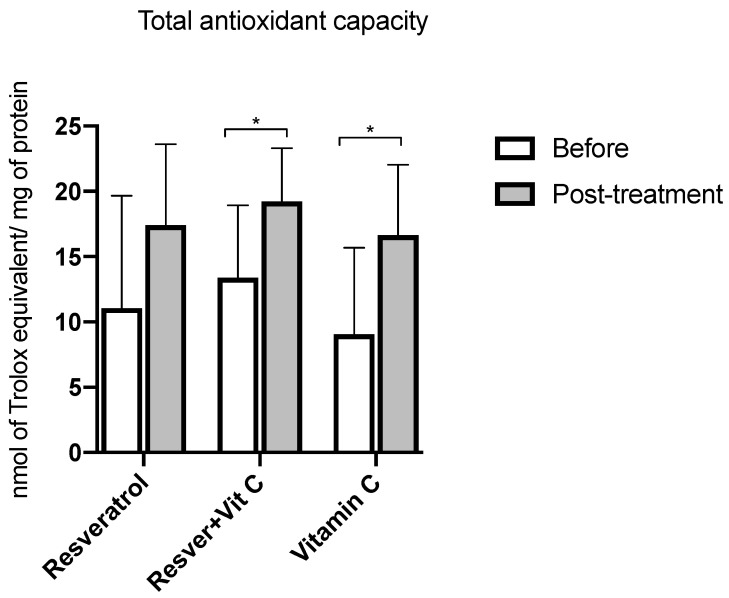
Total antioxidant capacity before and after treatment in postmenopausal women with insulin resistance. Resveratrol group, *n* = 13; resveratrol and vitamin C group, *n* = 15; and vitamin C group, *n* = 14. The results obtained were analyzed using Student’s “*t*” test. Data are presented as mean ± standard deviation. * *p* < 0.05.

**Figure 6 nutrients-16-03775-f006:**
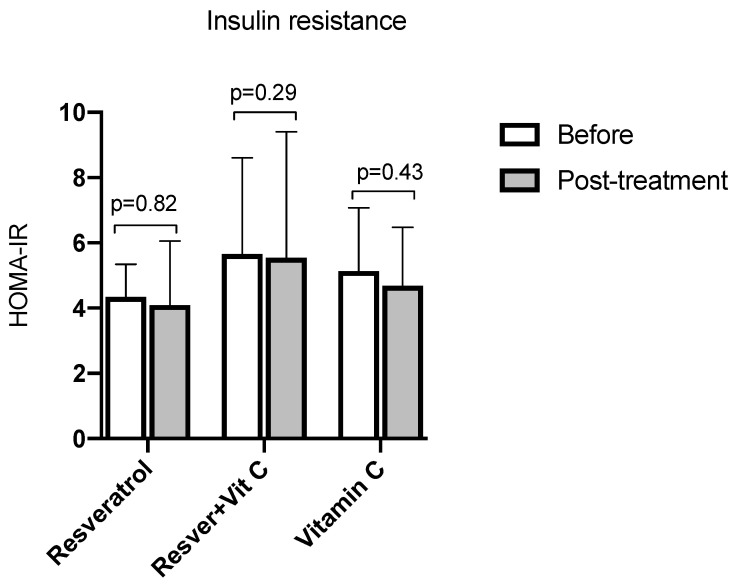
HOMA-IR before and after treatment in postmenopausal women with insulin resistance. Resveratrol group, *n* = 13; resveratrol and vitamin C group, *n* = 15; and vitamin C group, *n* = 14. The results obtained were analyzed using Student’s “t”. Data are presented as mean ± standard deviation.

**Table 1 nutrients-16-03775-t001:** Intra-group comparison of clinical and biochemical characteristics in postmenopausal women with insulin resistance at baseline and after three months of treatment.

	Resveratrol	Resveratrol + Vitamin C	Vitamin C
	(*n* = 13)	(*n* = 15)	(*n* = 14)
Basal			
Weight (kg)	69.2 ± 8.6	76.1 ± 10.4	79.4 ± 10.80
BMI (kg/m^2^)	29.4 ± 2.3	32.4 ± 4	32.8 ± 3.7
Glucose (mg/L)	105.9 ± 17.8	98.33 ± 99	96.5 ± 11.7
Insulin (μU/mL)	16.9 ± 2.8	22.14 ± 8.25	21.9 ± 9.4
HDL-C (mg/dL)	48.1 ± 20.4	50.3 ± 23.3	44.7 ± 6.1
Total cholesterol (mg/dL)	242.8 ± 81.5	217.1 ± 51.3	213.6 ± 32.7
LDL-C (mg/dL)	139.4 ± 43.8	132.8 ± 42.7	132.1 ± 35.4
Triglycerides (mg/dL)	208 ± 127.2	200.3 ± 66.4	223.2 ± 108.9
Uric acid (mg/dL)	5.4 ± 1.1	5.8 ± 1.4	5.9 ± 1.0
Post-treatment			
Weight (kg)	68.2 ± 8.5	74.9 ± 9.8	78.5 ± 9.60
BMI (kg/m^2^)	28.9 ± 2.1	31.7 ± 4	32.4 ± 3.50
Glucose (mg/L)	102.9 ± 15.8	92.27 ± 11.25	195.1 ± 9.9
Insulin (μU/mL)	16.5 ± 6.1	18.39 ± 7.08	19.9 ± 7.9
HDL-C (mg/dL)	51.8 ± 28.3	49.2 ± 23	40.8 ± 7.0
Total cholesterol (mg/dL)	236.1 ± 40.9 *	206.9 ± 39.6	205.3 ± 32.3
LDL-C (mg/dL)	150.9 ± 43	129.8± 30.6	121.8 ± 19.5
Triglycerides (mg/dL)	167.9 ± 73.3 *	171.4 ± 47.7 **	184.6 ± 59.9
Uric acid (mg/dL)	5.3 ± 1.1	5.5 ± 1.4	5.6 ± 1.2

BMI: body mass index; HDL-C: high-density lipoprotein cholesterol; LDL-C: low-density lipoprotein cholesterol. Data are presented as mean ± standard deviation. * *p* < 0.001 group A vs. C, ** *p* < 0.001 group B vs. C. One-way ANOVA test.

**Table 2 nutrients-16-03775-t002:** Inter-group comparison of clinical and biochemical characteristics in postmenopausal women with insulin resistance at baseline and after three months of treatment.

	Resveratrol (*n* = 13)			Resveratrol + Vitamin C (*n* = 15)			Vitamin C (*n* = 14)		
	Before	Post-Treatment	*p*-Value *	Before	Post-Treatment	*p*-Value *	Before	Post-Treatment	*p*-Value *
Age (years)	54.1 ± 3.2			55.2 ± 4			56.2 ± 4.2		
Weight (kg)	69.2 ± 8.6	68.2 ± 8.5	0.65	76.1 ± 10.4	74.9 ± 9.8	0.80	79.4 ± 10.8	78.5 ± 9.6	0.73
BMI (kg/m^2^)	29.4 ± 2.3	28.9 ± 2.1	0.85	32.4 ± 4	31.7 ± 4	0.35	32.8 ± 3.7	32.4 ± 3.5	0.55
Glucose (mg/L)	105.9 ± 17.8	102.9 ± 15.8	0.92	98.33 ± 99	92.27 ± 11.25	0.12	96.5 ± 11.7	95.1 ± 9.9	0.53
Insulin (μU/mL)	16.9 ± 2.8	16.5 ± 6.1	0.70	22.14 ± 8.25	18.39 ± 7.08	0.92	21.9 ± 9.4	19.9 ± 7.9	0.12
HDL-C (mg/dL)	48.1 ± 20.4	51.8 ± 28.3	0.79	50.3 ± 23.3	49.2 ± 23	0.62	44.7 ± 6.1	40.8 ± 7.0	0.50
Total cholesterol (mg/dL)	242.8 ± 81.5	236.1 ± 40.9	0.50	217.1 ± 51.3	206.9 ± 39.6	0.89	213.6 ± 32.7	205.3 ± 32.3	0.35
LDL-C (mg/dL)	139.4 ± 43.8	150.9 ± 43	0.33	132.8 ± 42.7	129.8± 30.6	0.34	132.1 ± 35.4	121.8 ± 19.5	0.25
Triglycerides (mg/dL)	208 ± 127.2	167.9 ± 73.3	0.83	200.3 ± 66.4	171.4 ± 47.7	0.85	223.2 ± 108.9	184.6 ± 59.9	0.40
Uric acid (mg/dL)	5.4 ± 1.1	5.3 ± 1.1	0.77	5.8 ± 1.4	5.5 ± 1.4	0.68	5.9 ± 1.0	5.6 ± 1.2	0.82

BMI: body mass index; HDL-C: high-density lipoprotein cholesterol; LDL-C: low-density lipoprotein cholesterol. Data are presented as mean ± standard deviation. * Paired *t*-test or Mann–Whitney U test.

## Data Availability

The datasets generated and/or analyzed during the present study are not publicly available due to privacy but are available from the corresponding author on reasonable request.

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
