# Peer review of "The Administration of Resveratrol and Vitamin C Reduces Oxidative Stress in Postmenopausal Women—A Pilot Randomized Clinical Trial"

_nutrients, 2024, doi:10.3390/nu16213775_

Round 1

Reviewer 1 Report

Comments and Suggestions for Authors

The manuscript prepared by Montoya-Estrada and colleagues describes the results of a pilot RCT intended to evaluate the effect of the combination of resveratrol and vitamin C in the levels of oxidative stress in postmenopausal women.

This is a relevant topic, as it is important to find the best strategies to deaccelerate the development of age-related diseases and reduce the excessive oxidative stress associated with ageing, specifically in women due to hormonal changes.

There are some concerns to be addressed.

The Introduction is globally well-written and addresses many important topics. I would recommend revising the English (ex.: line 99) and shorten some sentences (ex.: lines 50 to 56; lines 99 to 110).

Please also revise the references used (for instance, information from lines 85 to 92 is all from 2 refs only?).

Moreover, the text from lines 99 to 110 is not only very long for a solely sentence, as it is very confused, so authors should consider explaining better and better justify their hypothesis and aim based on this.

At the Materials and Methods, the authors should define the STRAW classification and explain how this classification was relevant for the study.

Please define: HOMA (IR or beta?), including the calculation and appropriate reference; altered lipid profile; altered glycemic profile.

Consider refining the description of inclusion/exclusion criteria overall.

Women should not be “eliminated”, rather "excluded from the study" (line 130).

Explain the meaning of sentence in lines 148-9.

Better describe the statistical analysis. Were these paired or unpaired? At the figure legends, indicate the n and the type of test used.

When starting the Results section, be aware that the text (page 4) does not reflect the data in Figure 1. Also, some information belongs to the Methods (dosage, time of administration; the information in line 220).

Please consider better explaining the results of Table 1 in the text and add the statistical analysis in the table (between groups at baseline and also paired).

Refer whether the baseline values of LHP, MDA, etc., are or not statistically different between the groups at baseline.

Be careful when considering that there was an effect (even if stated as a “trend”) in the HOMA-IR for the 3 groups, when the changes are very minimal in 2 groups (less than 10% change and p>0.05), and when in the combined group it shows high variability (and p>0.05). Otherwise, the authors would be also considering that that are no differences in the changes before and after in any of the obtained results in any of the groups, omitting the importance of statistical analyses, meaning that the results would be the same whether the women would take only resveratrol, only vitamin C or both.

The p value should be shown (Figure 6). Also, avoid the use of “notable decrease” (in line 341-2, at the Discussion), as it is for sure not notable.

The Discussion needs profound improvements. The authors should be clearer when stating their results (see my comment above regarding if there is enough evidence to state a better benefit of using the combination). Also, the authors should avoid trying to find other studies to validate their results (ex.: lines 312-3), otherwise what would be the point of performing the RCT?

Minor

Remove ERON from line 65, as in English the abbreviation normally used is ROS/RNS.

Legend for Figure 1 should be below (and not above) the figure.

Define “experimental subjects” (lines 286; 298-9).

Revise sentence “It was …” (line 297), as it does not make sense.

“more beneficial “ (line 305) in terms of what?

Define “mixed population” (line 307).

Add reference at the end of line 308.

Comments on the Quality of English Language

These were referred above in Comments and Suggestions for Authors.

Author Response

Reviewers' Comments to Author:

Reviewer: 1

Comments and Suggestions for Authors

The manuscript prepared by Montoya-Estrada and colleagues describes the results of a pilot RCT intended to evaluate the effect of the combination of resveratrol and vitamin C in the levels of oxidative stress in postmenopausal women.

This is a relevant topic, as it is important to find the best strategies to deaccelerate the development of age-related diseases and reduce the excessive oxidative stress associated with ageing, specifically in women due to hormonal changes.

There are some concerns to be addressed.

  1. The Introduction is globally well-written and addresses many important topics. I would recommend revising the English (ex.: line 99) and shorten some sentences (ex.: lines 50 to 56; lines 99 to 110).
  2. The paragraph was shortened.

  1. Please also revise the references used (for instance, information from lines 85 to 92 is all from 2 refs only?).
  2. We added three references.

  1. Moreover, the text from lines 99 to 110 is not only very long for a solely sentence, as it is very confused, so authors should consider explaining better and better justify their hypothesis and aim based on this.
  2. We rewrite the paragraph.

  1. At the Materials and Methods, the authors should define the STRAW classification and explain how this classification was relevant for the study.
  2. We added in methods the definition of STRAW classification and in the discussion section we explain how the classification was relevant for the study.

  1. Please define: HOMA (IR or beta?), including the calculation and appropriate reference; altered lipid profile; altered glycemic profile.
  2. HOMA-IR was defined, calculation and reference were added. Altered lipid profile and altered glucose were defined.

  1. Consider refining the description of inclusion/exclusion criteria overall.
  2. The inclusion and exclusion criteria was rewritten.

  1. Women should not be “eliminated”, rather "excluded from the study" (line 130).
  2. We perform the change according to suggestion.

  1. Explain the meaning of sentence in lines 148-9.
  2. The paragraph was rewritten.

  1. Better describe the statistical analysis. Were these paired or unpaired? At the figure legends, indicate the n and the type of test used.
  2. The statistical analysis was rewritten and added to figure legends the n and type of test used.

  1. When starting the Results section, be aware that the text (page 4) does not reflect the data in Figure 1.

R: We added the correct information.

  1. Also, some information belongs to the Methods (dosage, time of administration; the information in line 220).
  2. We agree, the information was moved to the methods section.

  1. Please consider better explaining the results of Table 1 in the text and add the statistical analysis in the table (between groups at baseline and also paired).
  2. We changed table 1 and in the new modified manuscript is table 2, we added table 1 to explain the comparison among groups baseline and three months after the intervention. We clarify the explanation of table 1 and 2.

  1. Refer whether the baseline values of LHP, MDA, etc., are or not statistically different between the groups at baseline.
  2. The p value was added between before and after three months of the intervention. We confirmed that there are no statistically significant differences among groups in baseline and three months after the intervention in LHP, MDA carbonyls and TAC

  1. Be careful when considering that there was an effect (even if stated as a “trend”) in the HOMA-IR for the 3 groups, when the changes are very minimal in 2 groups (less than 10% change and p>0.05), and when in the combined group it shows high variability (and p>0.05).
  2. We agree, we eliminated the term trend in all the results and discussion section.

  1. Otherwise, the authors would be also considering that that are no differences in the changes before and after in any of the obtained results in any of the groups, omitting the importance of statistical analyses, meaning that the results would be the same whether the women would take only resveratrol, only vitamin C or both.
  2. We agree, we clarify it in the results and discussion section, according with the statistical analyses.

  1. The p value should be shown (Figure 6). Also, avoid the use of “notable decrease” (in line 341-2, at the Discussion), as it is for sure not notable.
  2. We added the p value for figure 6 and we avoid the use of “notable decrease”.

  1. The Discussion needs profound improvements. The authors should be clearer when stating their results (see my comment above regarding if there is enough evidence to state a better benefit of using the combination).
  2. The discussion section was re-structured considering the comments of reviewers.

  1. Also, the authors should avoid trying to find other studies to validate their results (ex.: lines 312-3), otherwise what would be the point of performing the RCT?
  2. We agree, the discussion was modified.

Minor

  1. Remove ERON from line 65, as in English the abbreviation normally used is ROS/RNS.
  2. The abbreviations ROS/RNS have been corrected.

  1. Legend for Figure 1 should be below (and not above) the figure.
  2. We leave the title arrangement according to the instructions for the authors according to the journal

  1. Define “experimental subjects” (lines 286; 298-9).
  2. “Experimental subjects” has been defined in the text.

  1. Revise sentence “It was …” (line 297), as it does not make sense.
  2. The sentence was deleted

  1. “more beneficial “ (line 305) in terms of what?
  2. The sentence was completed with "by reducing the levels of thiobarbituric acid".

  1. Define “mixed population” (line 307).
  2. “Mixed population” has been defined, referring to men and women

  1. Add reference at the end of line 308.
  2. The reference has been added.

Reviewer 2 Report

Comments and Suggestions for Authors

The manuscript focuses on the administration of resveratrol and vitamin C and their capacity to reduce oxidative stress in postmenopausal women. In my opinion, this is an important study, especially due to the beneficial properties and the lack of toxicity at the recommended daily dose.

I have only small concerns:

1.       Correct the punctuation errors, example in lines: 2, 58, 77, 273, 300.

2.       Please explained all abbreviations used in the manuscript, example GLUT-4, Nrf2, ROS, SIRT1, DM2, CAT and other. While abbrevaiations of OS or MDA are explained more the once.

3.       Line 65 - in my opinion the abbreviation of reactive oxygen and nitrogen species is  RONS instead ERON.

4.       The sentence presents in lines 99-110 is too long.

5.       Please include the exclusion criteria in one paragraph. Lines 123-125 also describe exclusion criteria.

6.       Table 1 – How authors can explain the increased concentration of LDL after 3 months of resveratrol treatment? Please add the information about statistically significant differences. Correct the name “Colesterol (mg/dl)”

7.       Why is the unit of lipohydroperoxides expressed in mg dry weight when it was determined in plasma?

8.       Please correct the sentence (line 181-182) – “Because this is a pilot study, a convenience sample of 13 participants per group was  selected” – however, only group A consists of 13 participants.

9.       In line 279-280 “…Therefore, the use of antioxidants can be beneficial in reducing these markers.” Please extend the information about “these markers”.

10.   Lines 321-323 – because the authors did not assay the activity of CAT, they cannot claim that the the results of their study contradict those obtained by Amini et al.

11.   Line 298: “…Lin [35] and Kong [36]..” are not sole authors of the publication, therefore, it should be “Lin et al. [35] and Kong et al.  [36]”

12.   Line 368 correct:  HOMA-IR instead HOMA

Author Response

Reviewer: 2

Comments and Suggestions for Authors

The manuscript focuses on the administration of resveratrol and vitamin C and their capacity to reduce oxidative stress in postmenopausal women. In my opinion, this is an important study, especially due to the beneficial properties and the lack of toxicity at the recommended daily dose.

I have only small concerns:

  1. Correct the punctuation errors, example in lines: 2, 58, 77, 273, 300.
  2. Punctuation errors have been corrected in the text

  1. Please explained all abbreviations used in the manuscript, example GLUT-4, Nrf2, ROS, SIRT1, DM2, CAT and other. While abbreviations of OS or MDA are explained more the once.
  2. The abbreviations have been completed in the text and CAT has been changed to TAC

  1. Line 65 - in my opinion the abbreviation of reactive oxygen and nitrogen species is RONS instead ERON.
  2. We agree, it has already been modified in the tex.t

  1. The sentence presents in lines 99-110 is too long.
  2. We have modified and shortened the sentence.

  1. Please include the exclusion criteria in one paragraph. Lines 123-125 also describe exclusion criteria.
  2. The exclusion criteria were included in a single paragraph, according to your suggestion.

  1. Table 1 – How authors can explain the increased concentration of LDL after 3 months of resveratrol treatment? Please add the information about statistically significant differences. Correct the name “Colesterol (mg/dl)”
  2. Apparently an increase in LDL concentration is observed after 3 months of treatment, however; there are no statistically significant differences. The p values ​​have been added. The name of cholesterol was corrected.

  1. Why is the unit of lipohydroperoxides expressed in mg dry weight when it was determined in plasma?
  2. Another way besides expressing the results by mg of plasma, in the laboratory we have concluded that it is more precise to report the data by mg of organic matter due to all the organic components that the plasma contains such as plasma proteins, albumin, globulins and fibrinogen.

  1. Please correct the sentence (line 181-182) – “Because this is a pilot study, a convenience sample of 13 participants per group was  selected” – however, only group A consists of 13 participants.
  2. The sentence was corrected to fifteen participants per group. However, there were losses to follow-up.

  1. In line 279-280 “…Therefore, the use of antioxidants can be beneficial in reducing these markers.” Please extend the information about “these markers”.
  2. The idea in the text has been completed.

  1. Lines 321-323 – because the authors did not assay the activity of CAT, they cannot claim that the the results of their study contradict those obtained by Amini et al.
  2. There was an error, the abbreviation CAT is incorrect, it refers to TAC.

  1. Line 298: “…Lin [35] and Kong [36]..” are not sole authors of the publication, therefore, it should be “Lin et al. [35] and Kong et al.  [36]”
  2. The text has been corrected.

  1. Line 368 correct:  HOMA-IR instead HOMA
  2. The paragraph was corrected

Reviewer 3 Report

Comments and Suggestions for Authors

The manuscript reports interesting data on the effect of resveratrol and vitamin C, alone and in combination, on the markers of oxidative stress in postmenopausal women. In spite of a lower number of persons studied, the results are of interest and potential usefulness.

Remarks

In the Introduction, the authors seemingly postulate that the synergic action of vitamin C consists in preventing glutathione  oxidation by oxidized resveratrol. Such a mechanism would be operative in a cel-free system in vitro but in vivo is highly improbable as glutathione pool is much higher than the pool of introduced vitamin C and there is an efficient system of regeneration of GSH. It could be true in blood plasma but resveratrol readily penetrates the cel membrane and is expected to be located mostly in the cells.

The doses of the antioxidants taken daily should  be clearly stated; I guess that it was 500 mg resveratrol and 500 mg vitamin C daily. Am I right?

Was the diet of participants controlled? Some of them could take high doses of antioxidants with food.

Methods should cover all parameters studied, not only markers of oxidative stress, taking of the blood and possible storage of blood plasma etc.

A placebo group is mentioned (Lines 133-140) but this group does not appear in Results.

Results: The statistical significance of differences between values before and after the treatments are indicated but what with the statistical significance of differences between the treated groups. Are there any significant differences which could substantiate better action of the combination of antioxidants? Otherwise, there is no ground for such a statement.

Discussion:   Lines 284-286: „the combination of resveratrol and vitamin C is greater (33%) than when administered individually; resveratrol (25%) or vitamin C (15%)”. This statement is in disgreement with data shown in Figure 2.

The values of TAC reported in Figure 5 do not seem reliable. In the control, these values are about 1 nmol/mg protein. There is about 70 g  protein in 1 liter of blood plasma, so TAC would be about 0.07 mmol Trolox equivalents/L while most authors reported values about 1 mmol Trolox equivalents/L.

In summary, it seems that the manuscript escaped final edition, which would eliminate inconsistencies. Linguistic amendment would be desirable.

Other remarks

Line 77: „flavonoid; In its structure” please separate the sentences or change to „in”

Line 85: „Increases the activity”, this sentence has no subject

Line 90: „eicosanoid pro-inflammatory”, pro-inflammatory eicosanoids?

Lines 97/98:” The effects of resveratrol regulate insulin resistance and blood”, what in blood?

Lines 109/110:” such as deoxyribonucleic acid, lipids, and proteins”, it would be more appropriate to state „nucleic acids…”, there is much more RNA than DNA in the cells, and RNA is more easily available than DNA

Lines 181/182: the declared size of the groups (13) is in disagreement with the sizes stated in Lines 192/193 and with Figure 1. Next problem with the numer of participants: 13+15+14 is not 46 (Lines 191-193)

Figure 1: Perhaps „declined” rather than „decline”

Table 1: Please change „Kg” to”kg” and „dl” to”dL”, „Colesterol” to „Cholesterol”

Line 317: Please explain „CAT”.Is it an erroneous form of „TAC”?

Comments on the Quality of English Language

The text requires mainly grammatical check

Author Response

Reviewer: 3

Comments and Suggestions for Authors

The manuscript reports interesting data on the effect of resveratrol and vitamin C, alone and in combination, on the markers of oxidative stress in postmenopausal women. In spite of a lower number of persons studied, the results are of interest and potential usefulness.

Remarks

  1. In the Introduction, the authors seemingly postulate that the synergic action of vitamin C consists in preventing glutathione oxidation by oxidized resveratrol. Such a mechanism would be operative in a cell-free system in vitro but in vivo is highly improbable as glutathione pool is much higher than the pool of introduced vitamin C and there is an efficient system of regeneration of GSH. It could be true in blood plasma, but resveratrol readily penetrates the cel membrane and is expected to be located mostly in the cells.
  2. We agree with your comment and have therefore modified the text.

  1. The doses of the antioxidants taken daily should be clearly stated; I guess that it was 500 mg resveratrol and 500 mg vitamin C daily. Am I right?
  2. Correct, the doses were clarify in the last paragraph of Randomization and allocation concealment

  1. Was the diet of participants controlled? Some of them could take high doses of antioxidants with food.
  2. We added the recommendation indicated to the participants during the study.

  1. Methods should cover all parameters studied, not only markers of oxidative stress, taking of the blood and possible storage of blood plasma etc.
  2. We have completed the methods section.

  1. A placebo group is mentioned (Lines 133-140) but this group does not appear in Results.
  2. I appreciate your comment, in the first paragraph of Randomization and allocation concealment it mentions how the groups are formed with their placebo.

  1. Results: The statistical significance of differences between values before and after the treatments are indicated but what with the statistical significance of differences between the treated groups. Are there any significant differences which could substantiate better action of the combination of antioxidants? Otherwise, there is no ground for such a statement.
  2. We added a paragraph in the discussion section.

  1. Discussion:   Lines 284-286: „the combination of resveratrol and vitamin C is greater (33%) than when administered individually; resveratrol (25%) or vitamin C (15%)”. This statement is in disgreement with data shown in Figure 2.
  2. We corrected the sentence.

  1. The values of TAC reported in Figure 5 do not seem reliable. In the control, these values are about 1 nmol/mg protein. There is about 70 g  protein in 1 liter of blood plasma, so TAC would be about 0.07 mmol Trolox equivalents/L while most authors reported values about 1 mmol Trolox equivalents/L.
  2. We agree. There was a mistake in the conversion to nmol/mg protein. The correct data were added to the revised manuscript.

C.In summary, it seems that the manuscript escaped final edition, which would eliminate inconsistencies. Linguistic amendment would be desirable.

  1. We edited all inconsistences in the revised manuscript.

Other remarks

  1. Line 77: „flavonoid; In its structure” please separate the sentences or change to „in”
  2. The sentence structure was corrected

  1. Line 85: „Increases the activity”, this sentence has no subject
  2. Thanks a lot; we added the subject.

  1. Line 90: „eicosanoid pro-inflammatory”, pro-inflammatory eicosanoids?
  2. Text was corrected “Eicosanoids are those that regulate proinflammatory processes”.

  1. Lines 97/98:” The effects of resveratrol regulate insulin resistance and blood”, what in blood?
  2. The sentence was corrected in the text.

  1. Lines 109/110:” such as deoxyribonucleic acid, lipids, and proteins”, it would be more appropriate to state „nucleic acids…”, there is much more RNA than DNA in the cells, and RNA is more easily available than DNA
  2. That paragraph was shortened.

  1. Lines 181/182: the declared size of the groups (13) is in disagreement with the sizes stated in Lines 192/193 and with Figure 1. Next problem with the number of participants: 13+15+14 is not 46 (Lines 191-193)
  2. We corrected the error of the number of participants in the results section.

  1. Figure 1: Perhaps „declined” rather than „decline”
  2. Figure 1 was corrected.

  1. Table 1: Please change „Kg” to”kg” and „dl” to”dL”, „Colesterol” to „Cholesterol”
  2. It has already been addressed in the text.
  3. Line 317: Please explain „CAT”.Is it an erroneous form of „TAC”?
  4. Thank, the abbreviation has been corrected.

Round 2

Reviewer 1 Report

Comments and Suggestions for Authors

The authors have mostly consider and successfully respond to the concerns and suggestions, leading to an improvement of the manuscript.

Only a few minor issues:

Since there is a group with 21 individuals, in line 199 it should state "a minimum of 15".

Lines 318-22 are a repetition of the above paragraph.

Author Response

Reviewer: 1

Comments and Suggestions for Authors

The authors have mostly consider and successfully respond to the concerns and suggestions, leading to an improvement of the manuscript.

Only a few minor issues:

  1. Since there is a group with 21 individuals, in line 199 it should state "a minimum of 15".
  2. The change was made according to your suggestion

  1. Lines 318-22 are a repetition of the above paragraph.
  2. Thank you for your comment, the repeated paragraph was removed.
